# Improving Nature Connectedness in Adults: A Meta-Analysis, Review and Agenda

**David Sheffield \*** , **Carly W. Butler and Miles Richardson**

School of Psychology, University of Derby, Derby DE22 1GB, UK
* Correspondence: d.sheffield@derby.ac.uk; Tel.: +44-(0)-7825904935

**Abstract:** With clear links between an individual's sense of nature connectedness, their psychological wellbeing, and engagement in nature-friendly behaviours, efforts to improve people's relationships with nature can help unite human and planetary wellbeing. In the context of a rapidly growing evidence base, this paper updates previous meta-analytic reviews to explore the impact of (quasi-) experimental manipulations and field interventions on nature connectedness in adult populations. The analysis examines the relative effects of type of contact (direct or indirect), quality of engagement (active or passive) and the timing of the engagement (single session, repeated practice or residential). Results show a medium positive short-term mean effect of manipulations on nature connectedness, with similar effect sizes for immediate and sustained increases. No effect size differences were observed between different types of contact, quality, or timing of engagement. Follow-up measures were mostly used in studies involving regular and repeated practices. An agenda for future research and practice is put forward, emphasising the need for examining a wider range of nature engagement activities, greater understanding of factors leading to increases in nature connectedness, design and testing of practices for sustained nature connection, and initiatives that engage people with nature, create conditions for nature connection, and encourage repeated nature engagement activities.

**Keywords:** nature connectedness; nature relatedness; interventions; sustained change; meta-analysis

## 1. Introduction

There is global recognition that fostering closer connections with nature offers a solution to the joint issues of nature degradation, climate change, and human illbeing. Calls for restoring the human–nature relationship to address the nature and climate emergencies are growing in number and volume, with organisations like the United Nations [1] and World Economic Forum [2] highlighting the urgency and necessity of fixing our relationship with nature. While many solutions will target deeper leverage points at the national and global level, there is a need for parallel efforts to help individuals and communities feel closer to nature to create environments for thriving.

The subjective sense of feeling closer to nature is a key element of nature connectedness—a psychological construct that reflects how people think about, feel about, and relate to nature. Nature connectedness provides a useful and measurable focus in efforts to renew the human–nature relationship and is noted as a key realm for transformational sustainability interventions [3,4]. The applied potential of nature connectedness interventions to improve the human–nature relationship through applying the pathways to nature connectedness [5,6] is supported by recent evidence reviews [7]. This interest in nature connectedness stems from a strong and robust link with pro-environmental behaviours, pro-nature conservation behaviour [8,9], and a greater sense of well-being to levels above accepted benchmarks such as socioeconomic status [10–12]. Research also suggests a distinction between passive nature contact and nature connectedness, with nature connectedness being more important than number of visits to nature or time in nature in predicting wellbeing and engagement in pro-environmental behaviours [11,13].

In addition to cross-sectional research exploring the benefits of nature connectedness to human and nature's wellbeing, researchers have undertaken empirical work and developed interventions to improve nature connectedness. These interventions often target nature connectedness through active engagement with nature as a route to improving mental health and wellbeing and have successfully delivered sustained improvements in these outcomes through active and direct [14] and indirect engagement with nature [15]. Some interventions are designed to activate pathways to nature connectedness [5], and it is suggested that active and direct engagement with nature explains levels of nature connectedness [16]. Given the benefits to wellbeing and the urgent need for greater pro-nature and environmental actions, greater understanding of how nature connectedness develops and how it can be improved is crucial.

Three recent literature reviews have sought to examine the factors related to nature connectedness. Barragan-Jason et al.'s meta-analytic review [17] examined correlational (n = 147) and experimental (n = 59) studies on human nature connectedness, with a total sample size of 70,523. Cross-sectional studies showed that human–nature connection is positively correlated with naturalist knowledge, time spent outside, engagement in mindfulness practices, pro-environmental and humanistic values, happiness and good health; and negatively correlated with materialism/consumerism and political conservatism. Analysis of results from experimental studies suggested interventions and manipulations can successfully enhance nature connectedness, particularly interventions involving direct contact with nature and what they describe as mindfulness practices: "focusing one's attention on one's inner self and one's environment in the present moment" (p. 3). They also found that while both short and longer interventions enhanced nature connectedness immediately post intervention only longer interventions resulted in longer term improvements (≥2 weeks). However, in the context of building a new relationship with nature there is a need to highlight interventions that can deliver sustained benefits before reaching conclusions on the best approaches to delivering improved nature connectedness. Here, Barragan-Jason et al. make a valuable contribution by showing that the estimated effects of environmental education, an assumed solution, were found to be minimal.

Lengieza and Swim's qualitative review of psychological literature on the antecedents of connectedness to nature [18] identified 85 cross-sectional and experimental papers. Discussion of these was organised into three areas: situational contexts, individual differences, and internal psychological states. They identify a need for additional theories about how connectedness is formed and call for more focus on process, and greater differentiation between similar connectedness antecedents. While contact with nature is strongly associated with connection, there has been limited research examining when contact with nature does not lead to connectedness or even results in decreased connectedness. The review identifies a need to understand what types of nature contact matter for growing connections, and whether different activities in nature are more effective in developing nature connectedness. While some contact with nature involves noticing nature (e.g., [19]), in other cases nature is more of an arena for other sorts of activities, "becoming a non-salient background element of the experience" (p. 8). Lengieza and Swim ask whether activities that 'enhance' nature contact bring something additional and lead to effects above and beyond those gained from simple nature contact.

Barrable and Booth reviewed literature on interventions designed to increase the nature connectedness of children and young people under the age of 18 [20]. They identified 14 papers meeting their criteria, which included the use of experimental or quasi-experimental design, nature connection as dependent variable, and use of a validated scale to measure nature connectedness. The length and form of the interventions varied widely—from two-hour field trips to programs that ran over several weeks. Nine of the studies identified themselves as involving environmental education, while others included camps/residentials, leisure and educational activities, and expeditions/field trips. While most activities had a heavy educational component, three studies reported the value of play and creativity in supporting nature connection. Barrable and Booth note the lack of research

into non-educational interventions and different ways of engaging with nature such as those seen in forest schools, nature kindergartens and so on. They also note limitations in terms of research design (e.g., few studies use control groups) and reporting.

Although there has been a surge of interest and research into nature connectedness, work on targeted interventions to deliver sustained improvements is at a relatively early stage. Thus, there is a need to identify the gaps in our knowledge as well as identifying promising approaches for further research. Building on earlier reviews and analyses, this paper examines experimental and quasi-experimental research that measures the impact of nature contact and engagement on nature connectedness amongst adults. In contrast to the broader scope of Barragan-Jason et al. [17] and Lengieza and Swim [18], our focus is on studies involving forms of nature contact and engagement that could be applied by individuals and organisations looking to increase nature connectedness—whether for preventative or therapeutic health and wellbeing initiatives, public health campaigns, environmental engagement and awareness raising, or more generally in efforts to reconcile the wellbeing of people and planet. To explore the qualities of various approaches to improving nature connectedness systematically, we examine the relative effects of type of nature contact (direct or indirect), quality of nature engagement (passive or active), and the number and nature of sessions involved in the interventions (one-off, repeated, or residential). A more focused review and meta-analysis offers a resource for those looking to contribute new knowledge in this area, as well as identifying approaches that work for interventions across different scales. To this end, we offer an agenda for research and practice based on the conclusions that can be drawn from the extant literature.

## 2. Methods

### 2.1. Inclusion Criteria

A literature search was carried out with the following inclusion criteria: (a) Peer-reviewed publications written in English language; (b) Experimental or quasi-experimental design with results for pre- and post-testing reported; (c) Nature connectedness as an outcome variable, measured using established nature connection scales (Inclusion of Nature in Self (INS) [21]; Extended Inclusion of Nature in Self (EINS) [22]; Nature Relatedness Scale (NR) [23]; Nature Relatedness—Short Form (NR6) [24]; Connectedness to Nature Scale (CNS) [25]; Implicit Association Test (IAT) [26]; Nature Connection Index (NCI) [27]; (d) Involving contact with nature via actual nature (e.g., walking outdoors), images of nature (e.g., photos, videos, or immersive virtual reality), or mental/imagined contact (e.g., guided imagery, meditation); (e) Majority of participants over the age of 18 years old.

### 2.2. Search Strategy

Searches were initially carried out using PsycInfo and Web of Science, using the keywords "nature connect*," "nature relatedness," "connection to nature," "connection with nature". Searches were restricted to peer-reviewed works published in English, with adult participants, published or available as a pre-print before 1 June 2022. Papers cited by Barrable-Jason et al. [17], and Lengieza and Swim [18] were also reviewed, along with papers in the authors' collections. Google scholar was used to identify papers who cited the works reviewed. Duplications and exclusions were removed, and full papers reviewed against the inclusion criteria. Questions about inclusion of certain papers were resolved through discussion amongst co-authors. The final collection included 36 papers and 48 effects (see Figure 1) (see Appendix A for PRISMA Flowchart).

To explore factors that impact on the effectiveness of contact and engagement with nature on nature connectedness, we coded studies on the following dimensions:

1. Type of contact: Direct vs. Indirect

The distinction is usually made between contact with real versus virtual nature, presumably reflecting the use of virtual reality technology for immersive nature experiences in contrast to contact with actual nature. We have instead distinguished between *direct* and *indirect* contact with nature to put the emphasis on the type of contact or engagement

an individual has with nature rather than the ontological status of the stimulus or means of delivery. Direct contact involves seeing or hearing actual nature in real-time. Indirect contact involves mediation of nature through audio-visual presentation or devices (e.g., Virtual Reality (VR) headsets, computer screen or posters), or imagined contact with nature. This distinction better captures activities that involve use of mental imagery (e.g., [15,28]), and use of nature images (e.g., [29]).

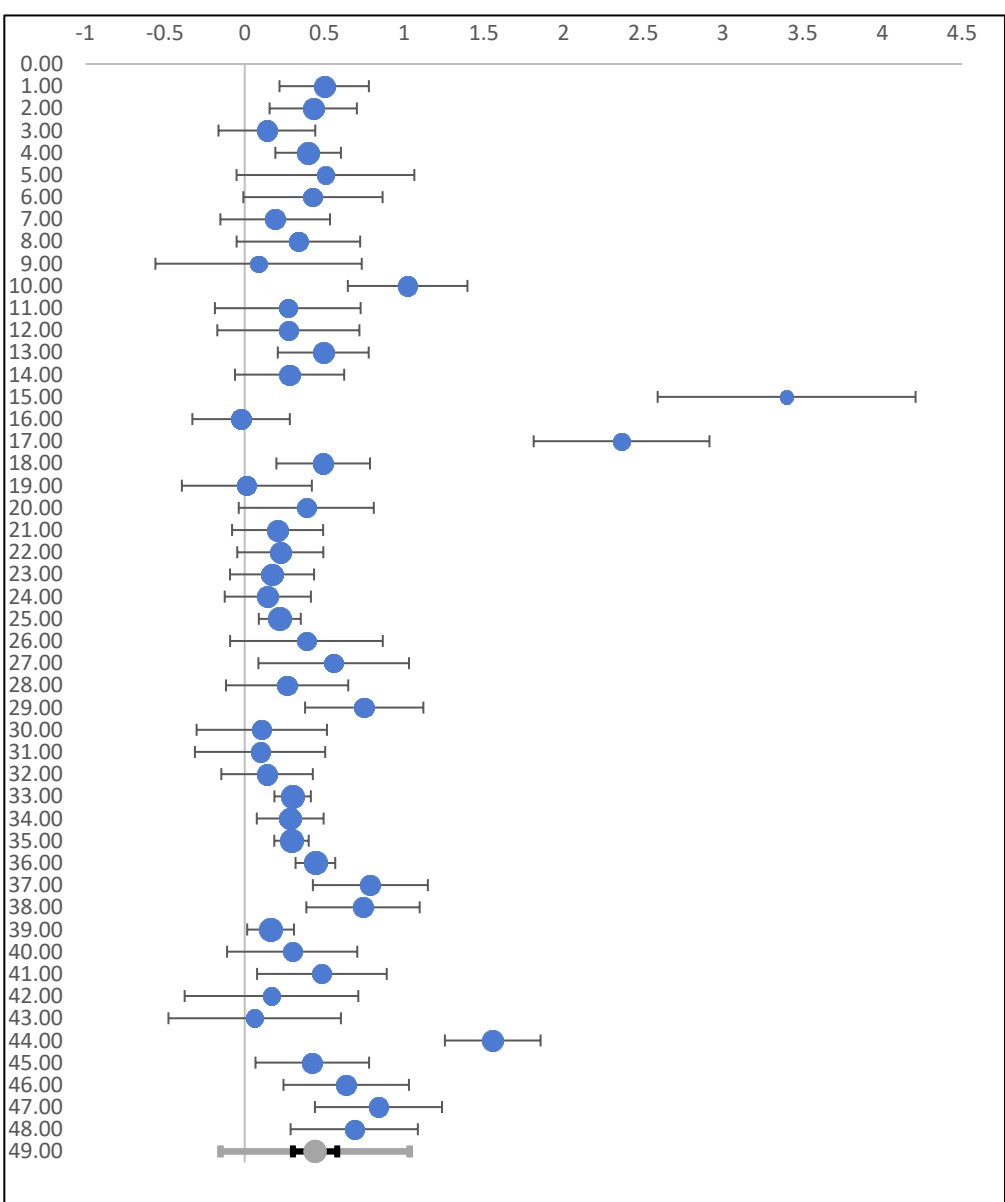

**Figure 1.** Forest Plot of Pre-Post Effects and Confidence Intervals.

2. Quality of engagement: Active vs. passive

We categorised studies based on the reported instructions given to participants about their contact or engagement with nature. Following [16], *active* (or psychological) engagement was identified in studies where participants were invited to be aware of or appreciate nature—whether through paying attention to sensory qualities (actual or imagined), nature's beauty, meanings, or emotional impact [5]. Studies were coded as involving *passive* engagement where no instructions about how to engage with nature were given to participants. This included studies where participants were asked to go on a 10 min walk, or to watch a series of photos of nature, or residential experiences where no explicit

psychological engagement instructions were reported. Our focus is on the reported study methodology, rather than participants' actual experiences. This allows us to explore the relative impact of activities and interventions where there is an explicit instruction to make psychological connections with nature, compared to when people are asked simply to be in, or look at, nature for a period of time. The distinction between connection or contact in nature experiences is crucial in aiding understanding about how best to foster closer relationships with nature.

3.  Timing and nature of activity (residential, single, or repeated engagement)

Studies were categorised in terms of the length and patterning of the nature contact involved, falling into one of three main groupings. *Residentials* involved at least one day and night. *Single* involved a discrete nature activity. *Repeated* included activities that were carried out for shorter periods over multiple days, whether daily, weekly, or several times over a set period.

## 3. Results

### *3.1. Characteristics of Included Studies*

#### 3.1.1. Publication Trends

There has been a marked acceleration of research in this area, with ten papers published between 2007 and 2017, and 26 published between 2017 and 2022. This increased rate of publication illustrates the establishment and acceptance of nature connectedness as a psychological construct with reliable and validated scales for its measurement, and the growing recognition of both the possibility and the value of increasing nature connectedness for human and planetary wellbeing; see Tables 1 and 2.

Research is dominated by studies carried out in Europe (n = 20) and North America (n = 9), with just three studies from Oceania, two from Asia and one from Africa. In terms of measures, the most widely used scale was the INS (n = 16), followed by the CNS (n = 15), the NR6 (n = 5), NR (n = 4), NCI (n = 2), EINS (n = 1), IAT (n = 1); some studies used more than one scale.

**Table 1.** Studies involving direct contact and engagement with nature.

| Citation | Country | Length | Activity | Engagement | Design | Follow-Up | n | Scale | Results | Other Measures |
|---|---|---|---|---|---|---|---|---|---|---|
| Barrable and Booth (2020) [30] | Scotland | 10 m Single | Group walk around campus. Notice nature's beauty and either (a) mentally note or (b) use phone to capture three beautiful things. | Active | Treatment comparison | 3 w (n = 11) | 57 | CNS | Increased NC in both conditions, no difference between conditions | Feelings, Nature noticing |
| Barrable and Lakin (2020) [31] | Scotland | 3 h Single | Student teachers exploring botanic garden, pond-dipping as part of research into pond diversity and composition. | Active | Pre-post | No | 49 | NR | Increased NC | Competence and willingness to teach outside |
| Cervinka et al. (2020) [32] | Austria | 2.5 h Single | Guided forest tour (groups of 2–7) with four stopping points. Sit or lie down and explore each spot with all senses for ten minutes. | Active | Pre-post | No | 99 | INS | Increased NC | Mood, Stress, Restoration, Mindfulness, Qualities of places |
| Choe et al. (2020a) [33] | UK | 1 h × 6 w Repeated | Mindfulness-Based Stress Reduction (MBSR) in natural outdoor environment | Passive | Treatment comparison (a) nature (b) built (c) indoor | 1 month | 99 | NR6 | NC increased in nature but not built or indoor environment. | Mindfulness, Mood, Eudaimonic wellbeing, Depression, Anxiety |
| Chou and Hung (2021) [34] | Taiwan | 30 m × 8 w Repeated | Participants asked to walk a forest trail on campus once a week for 8 weeks | Passive | Pre-mid-post | No | 10 | NR | Increased NC after 8 weeks, but not after 4 weeks. | Mental health, Learning engagement, Attention recovery and reflection |
| Deringer et al. (2020) [35] | United States | 4 d Residential | Backpacking trips in mountains. Mindfulness exercises each morning. | Passive | Control—no outdoor activity | No | 37 | INS CNS | Increased NC at mid- and post-trip compared with control | Ecological behaviour Mindfulness |
| Djernis et al. (2021) [36] | Denmark | 5 d Residential | Residential mindfulness (MBSR) programme in a therapy garden setting, inside an arboretum. | Passive | RCT, Treatment comparison (a) nature (b) indoors (c) control | 12 w | 60 | CNS | No group differences post-treatment, but at follow-up NC in outdoor group higher than control. | Stress, Self-compassion Mindfulness |
| Down et al. (2021) [37] | Australia | 3 d Residential | Outdoor expedition for pre-service teachers. | Passive | Pre-post | No | 54 | CNS | Increased NC | Wellbeing |
| Hamann and Ivtzan (2016) [38] | Multiple | 30 m × 30 d Repeated | 30 min in nature for 30 days. | Active | Control—waitlist | No | 62 | CNS | No increase in NC | Mood, Wellbeing Environmentally friendly behaviour, Meaning and spirituality, Mindfulness Nature contact |
| Johnson-Pynn et al. (2014) [39] | Uganda | 2–3 d Residential | Environmental education workshops with scientists, carrying out biodiversity assessments in rural or urban settings for two or three days. | Active | Treatment comparison—(a) 2 vs. (b) 3 days, rural vs. urban | No | 84 | INS CNS | Increased INS, higher for 3-day than 2-day. Decreased CNS for 2-day workshop, and urban workshop. | Self-efficacy, Civic attitudes and skills |

**Table 1.** *Cont.*

| Citation | Country | Length | Activity | Engagement | Design | Follow-Up | n | Scale | Results | Other Measures |
|---|---|---|---|---|---|---|---|---|---|---|
| Keenan et al. (2021) [40] | UK | 30 m × 5 days Repeated | Guided group walk, noticing three good things in nature. | Active | Treatment comparison (a) nature (b) urban | 6 w | 50 | CNS | Increased NC at post and follow-up | Wellbeing, Affect |
| Lim et al. (2020) [41] | Singapore | 2 h Single | Walk in biophilically designed hospital grounds (a) guided forest therapy walk in groups of 8 with sensory engagement and mindfulness activities (b) unguided with printed suggestions for sensory engagement with nature | Active | Treatment comparison (a) guided forest therapy (b) unguided sensory engagement | No | 51 | CNS | Increased NC both conditions, no difference between conditions | Environmental Identity Mood, Heart Rate |
| Lumber et al. (study 3) (2017) [5] | UK | 20 m Single | Guided group walk on campus with stops for pathways to nature connectedness activities: emotion-beauty, meaning-beauty, and compassion-beauty | Active | Treatment comparison (a) pathways activities (b) walk with no activities (c) walk in built environment | No | 72 | NRS | Increased NC for pathway activities, not built or nature control conditions. | Vitality, Physical Activity |
| Macaulay et al. (2022) [42] | Australia | 20 m Single | Time in nature with instructions for different ways of engaging with nature. | Active | Treatment comparison, (a) mindful engagement (b) directed engagement (c) mind wandering (d) unguided control (e) no instructions | No | 215 | CNS | No increase in NC for any condition | Mindfulness, Mood Attention |
| McEwan et al. (2019) [14] | UK | 7 d Repeated | Participants sent a prompt via a smartphone app to record 'one good thing in nature' when in a green space | Active | Treatment comparison (a) nature (b) built | 4 w (n = 164) | 322 | INS | Inc NC both groups at post- and follow-up. Stronger effect for nature condition | Quality of Life, Mood, Engagement with Beauty Nature exposure |
| McEwan et al. (2021) [43] | UK | 20 m Single | Guided group forest walks with forest bathing [FB]—explore with senses and/or compassionate mind training [CMT]—psychoeducation and guided imagery to inspire compassion for self, other humans, other species, and environment | Active | Treatment comparison (a) FB (b) CMT (c) FB + CMT | No | 61 | INS | Increased NC for FB and FB + CMT | Mood, Compassion, Pro-environmental attitudes, Rumination, Heart rate |
| McEwan, Richardson et al. (2021) [44] | UK | 30 d Repeated | Participants sent a prompt via a smartphone app to record 'one good thing' when in a green space | Active | Treatment comparison (a) nature (b) built | 12 w (n = 10) | 60 | NR6 INS | Increased NC for both nature and built condition at post. | Quality of life Positive affect Engagement with natural beauty |
| Nisbet and Zelenski (2011) (study 1) [45] | Canada | 17 m Single | Walk along a canal | Passive | Treatment comparison (a) outdoors (b) indoors | No | 150 | INS | NC higher in outdoor vs. indoor | Mood Relaxation Soft fascination |

**Table 1.** *Cont.*

| Citation | Country | Length | Activity | Engagement | Design | Follow-Up | n | Scale | Results | Other Measures |
|---|---|---|---|---|---|---|---|---|---|---|
| Passmore et al. (2022) [46] | Canada | 2 w Repeated | Ps asked to notice and be mindful of how natural elements and objects made them feel over the course of two weeks, and upload at least 10 photos of the scenes/objects that evoked emotions and written descriptions of emotions. | Active | Treatment comparison (a) nature (b) built (c) delay | No | 65 | INS CNS | Increased NC (INS) in nature condition but not built or delay conditions. | Positive and negative affect, Satisfaction with life, Meaning in life, Transcendent connectedness, Elevation, Hope |
| Richardson and McEwan (2018) [47] | UK | 30 d Repeated | Engage in 'wild activity' every day, choosing from activities designed to promote active engagement with nature | Active | Pre-post | 8 w | 308 | INS | Increased NC at post and follow-up | Engagement with beauty, Health, Happiness, Conservation behaviour, Emotion regulation |
| Richardson and Sheffield (2017) [48] | UK | 5 d Repeated | Notice and note down 'three good things in nature' every day. | Active | Treatment comparison (a) three good things in nature each day (b) three factual things each day | 8 w | 92 | CNS | Greater increase in NC for nature group at post and follow-up. | Health Linguistic Inquiry |
| Richardson et al. (2016) [49] | UK | 30 d Repeated | Engage in 'wild activity' every day, choosing from activities designed to promote active engagement with nature | Active | Pre-post | 12 w (n = 126) | 344 | INS, NCI | Increased NC at post and follow-up | Health Happiness Conservation behaviours |
| Richardson et al. (2018) [50] | UK | 30 d Repeated | Engage in 'wild activity' every day, choosing from activities designed to promote active engagement with nature | Active | Pre-post | 12 w (n = 273) | 655 | INS, NCI | Increased NC at post and follow-up | Health Happiness Conservation behaviours |
| Rogerson et al. (2020) [51] | UK | 12–20 m Single | 3 km run alone or in a group with 4–5 others in university sports fields (flat grass, views of trees and grassland, abundance of wildlife) | Passive | Treatment comparison (a) alone (b) group | No | 40 | CNS | Increased NC in both groups | Self esteem, Mood |
| Schultz and Tabanico (2007) (study 4) [52] | United States | 4–6 h Single | Visitors to San Diego Wild Animal Park | Passive | Pre-mid-post | No | 40 | IAT INS | Increased NC from entry to exit | Environmental concern Mood |
| Unsworth et al. (2016) [53] | United States | 3 d Residential | Meditate for 15 min in the morning while at nature camp. | Passive | Treatment comparison (a) meditation (b) no meditation | No | 71 | INS | Increased NC in meditation condition but not in no meditation condition. | Mindfulness |
| Warber et al. (2016) [54] | | 4 w Residential | National Youth Science Camp with lectures, hands-on studies, and outdoor adventure activities | Passive | Pre-post | No | 36 | CNS | Increased NC | Relationship with and experience of nature, Physical health, Psychological health, Emotional health, Social health, Spiritual health |

**Table 2.** Studies involving indirect contact and engagement with nature.

| Citation | Country | Length | Activity | Engagement | Control | Follow-Up | n | Scale | Results | Other Measures |
|---|---|---|---|---|---|---|---|---|---|---|
| Artbuthnott et al. (2014) (Study 3) [29] | Canada | 7 m Single | Look at 44 slides from the Natural History Museum, mostly featuring pictures of animals and plants. | Passive | Treatment comparison (a) nature images (b) built environment images | No | 56 | CNS | NC higher in nature vs. built condition | Wellbeing Pro-environment goals |
| Chan et al. (2021) (Study 1) [55] | Singapore | 5 m Single | Virtual reality walk in nature or urban setting, using headset. | Passive | Treatment comparison (a) nature (b) urban | No | 30 | CNS | NC increased in nature condition but no change in urban condition | Mood Cardiovascular activity, Prior VR experience |
| Choe (2020b) [56] | UK | 1 h × 3 w Repeated | MBSR or relaxation in simulated natural environment (room with images of nature on the walls) | Passive | Treatment comparison with 8 conditions: a0 MBSR or b0 relaxation with simulated nature (woodland or parkland) or non-nature (urban or empty room) | 1 w | 122 | NR6 | NC increased for relaxation in nature group but not MBSR in nature or non-nature conditions | Mindfulness, Mood, Depression, Anxiety, Environmental preference |
| Coughlan et al. (2022) [28] | Australia | 10 m Single | Guided imagery (GI)—taking a walk in natural setting, with emphasis on sensory imagery | Active | Treatment comparison (a) nature (b) urban (c) waitlist | No | 133 | EINS | Increased NC in nature GI but not other conditions. | Experiential ratings |
| Muneghina et al. (2021) [15] | UK | 10 m × 5 d Repeated | Nature based guided audio meditation with natural soundscape, designed to activate pathways to nature connectedness | Active | Treatment comparison (a) meditation (b) waitlist | 2 w | 72 | NR6 | Increased NC at post and follow-up | Anxiety, Paranoia, Mindfulness |
| Ray et al. (2021) [57] | United States | 15 m × 5 d/w × 4 w Repeated | Guided imagery audio with natural sounds | Passive | Treatment comparison (a) nature sounds (b) sounds of yoga/meditation class | No | 97 | CNS | NC increased in nature GI condition but not the class condition | Mindfulness, Pro-environmental behaviours |
| Sneed et al. (2021) [58] | | 10–15 m Single | Immersive 360-degree videos of nature reserve or library watched with headset vs. walk and observation in actual nature | Passive | Treatment comparison (a) virtual nature (b) virtual built (c) actual nature | No | 73 | NRS | Higher NC in direct nature condition vs. both indirect conditions. No difference between nature and built indirect. | State of interdependence with nature |
| Spangenberger et al. (2022) [59] | Germany | <7 m Single | Immersive virtual reality (iVR)—nature video shown from perspective of a tree | Passive | Treatment comparison (a) iVR (b) video on desktop | No | 28 | NR6 | No increase in NC for either condition | Perceived immersion, Perspective taking |
| Yeo et al. (2020) [60] | UK | 5 m Single | Watch virtual underwater coral reef | Passive | Treatment comparison (a) TV (b) 360-degree VR with head-mounted display or (c) interactive computer-generated VR (CG-VR) | No | 96 | INS | Increase in NC in conditions combined, greatest increase in CG-VR. | Presence, Boredom, Mood, Previous experience |

### 3.1.2. Type of Contact with Nature

Twenty-seven (75%) studies involved *direct contact* with nature, three times the number of studies exploring *indirect contact* with nature (n = 9). Indirect contact was facilitated through images (n = 2), videos (n = 3), audio (n = 1), or guided imagery/meditation practices (n = 3) involving imagining the sensory experience of being in nature. Of the studies involving direct contact, six (17%) were residential camps or experiences with participants taking part in a range of outdoor activities in a variety of settings. Ten studies involved walking in nature in forests, nature reserves, gardens, or urban nature, one involved running in a university sports field [51], and one involved visiting an animal park [52]. Five studies invited participants to carry out nature-based activities in their own time [38,42,47,49], while four invited participants to engage with nature in particular ways (i.e., appreciating and noticing it) during their daily lives without asking them to spend any extra time outside [14,44,46,48].

### 3.1.3. Quality of Contact and Engagement with Nature

There was a roughly even split between studies prompting for active (53%) or passive (47%) engagement with nature. Passive engagement involved exposure to nature, whether actual or mediated, without explicit instructions to engage with or experience it in particular ways. Most studies with indirect contact involved passive engagement, with participants asked to look at photos or videos [29,55,59,60] or where nature images or sounds featured as backgrounds to meditation [57] or mindfulness training [33].

Studies involving residential camps and activities were coded as passive engagement where there was no explicit mention of any activities based on sensory or psychological engagement with nature. While in practice such camps may have involved many opportunities for active engagement with nature, if not active encouragement to do so, we have coded studies based on the published methodologies. Only one paper involving a residential camp explicitly referred to activities that promote psychological engagement with nature [39].

Beyond the residential camps, passive engagement with nature was identified in six studies involving direct nature contact. This included studies where participants carried out an activity in a natural setting (i.e., MBSR training [56], visiting an animal park [52]), or were simply asked to walk [45,58] or run [51] outside.

Studies inviting participants to actively engage with nature by noticing, appreciating, or experiencing it in specified ways included those evaluating nature engagement campaigns such as 30 Days Wild in the UK [47,49,50], and an online nature-based intervention programme-based on the Suzuki 30 × 30 challenge [32]. These invited people to engage with nature daily, with suggestions of activities promoting nature connection. Four studies involved the 'three good things in nature' practice, with participants invited to appreciate nature's beauty and wonder over the course of one to two weeks [14,36,48,49]. One study asked people to notice when nature elicited strong emotions in them over the course of two weeks, and to describe and take photos of the scenes or features triggering these emotions [46]. Several studies invited people to deepen their engagement with nature through senses [32,41,43] and attention to nature's beauty [30]; through activation of multiple pathways to nature connection [5]; or by comparing the effects of different forms of mindful engagement with nature [57]. Two of these explicitly examine the impact of forest bathing techniques [41,43].

Two of the active engagement studies involved indirect contact with nature. These entailed guided imagery [28] and an audio-meditation [15] that invited participants to imagine sensory and emotional nature experiences. In these cases, there is clear psychological engagement with nature without actual physical contact with it.

### 3.1.4. Timing of Nature Contact and Engagement

Almost half of the studies (n = 16) examined the impact of brief one-off periods of contact or engagement with nature, which in most cases (n = 13) lasted twenty minutes

or less, with three studies involving two to three hours [31,38,41] and one between a half to one-day [52]. The short one-off activities included all studies of indirect nature contact except for [15]. In many cases, brief nature contact was used to compare different forms of contact and engagement, for instance, comparing direct and indirect contact [58], different digital and virtual reality technologies (e.g., [59,60]), walking inside compared to outside [5,45], or walking in built versus natural environments [5].

Six studies involved intensive residential experiences lasting from 2 days to 1 month. These covered a diverse range of nature engagement activities, including education [37,39,54], hiking [35], and meditation and mindfulness [35,36,53].

Fourteen studies involved participants having repeated (daily, weekly or several times over the course of one to eight weeks) nature contact or engagement, over periods ranging from 5 days to two months. These included daily nature contact or activities over the course of 30 days [38,47,49,50], noticing three good things in nature every day for a week or month [14,44,48], mindfulness and meditation-based practices [15,33,56,57], regular walking for eight weeks [60], and noticing emotional responses to nature over a two-week period [46].

### 3.1.5. Sustained Effects

Fourteen studies included follow-up measures between one and 12 weeks after completion of the study, though response rates were too low in two studies for inferential analysis [30,36]. Sustained benefits were observed in all studies that included follow-up measures, which included all studies involving repeated nature contact or engagement as discussed above, with the exception of [46,57]. One study involving residential nature engagement also measured sustained effects [36]. As above, lasting increases in nature connection were observed after regular nature activities and nature-noticing practices, as well as regular mindfulness and meditation practices carried out in real or simulated nature contexts.

### 3.2. Meta-Analyses

Meta-analyses were conducted focusing on pre-post and pre-follow-up comparisons. We used means and standard deviations (some of which were derived from standard errors or confidence intervals) to determine effect sizes for the interventions; for one study [45] we used data available from [17]. Data were entered into Meta-Essentials [61].

Variability was examined using Cochran's Q and $I^2$. Heterogeneity among effect sizes was determined by a significant Q value ($p < 0.10$). The $I^2$ statistic indicates the degree of variability in effect sizes (low heterogeneity, 1–49; moderate heterogeneity, 50–74; high heterogeneity, 75–100). In the case of significant heterogeneity, subgroup and moderator analyses were undertaken. Random effects models were used as the included studies included diverse manipulations; thus, heterogeneity was assumed (mean effect sizes = small, 0.10–0.29; moderate, 0.30–0.49; high, $\geq$0.50; Cohen [62]. Where there was high heterogeneity, three subgroup analyses were conducted to appraise whether (i) type of contact or engagement with nature (direct vs. indirect), (ii) quality of engagement (passive vs. active), (iii) timing of intervention (single vs. repeated vs. residential) accounted for the variability in the effects. Lastly, publication bias was examined using Funnel plots and associated statistics.

### 3.2.1. Immediate Effects of Interventions

The total sample size for interventions assessing nature connectedness pre and post intervention was 2855 (range 10–344) from 36 studies with 48 effects. Table 3 displays the effects of each intervention and Figure 1 shows the forest plot. There was a medium positive mean effect, g = 0.44 [95% CI 0.31, 0.58], which was significant ($p < 0.001$). Heterogeneity of effects was significant (Q = 262.48, $p < 0.001$) and inconsistency was high ($I^2$ = 82.09%).

**Table 3.** Immediate Effects and Confidence Intervals of Interventions along with Weighting.

| Study Name (Year) Condition | Hedges' g | Lower CI | Upper CI | Weight |
|---|---|---|---|---|
| Arbuthnott et al. (2014) [29] | 0.50 | 0.22 | 0.78 | 2.31% |
| Barrable and Booth (2020) [30] | 0.43 | 0.16 | 0.70 | 2.34% |
| Barrable and Lakin (2019) [31] | 0.14 | −0.16 | 0.44 | 2.25% |
| Cervinka et al. (2020) [32] | 0.40 | 0.19 | 0.60 | 2.54% |
| Chan et al. (2021) [55] | 0.51 | −0.05 | 1.06 | 1.57% |
| Choe et al. (2020a) [33] | 0.43 | −0.01 | 0.87 | 1.86% |
| Choe et al. (2020b)a [56] | 0.19 | −0.15 | 0.54 | 2.13% |
| Choe et al. (2020b)b [56] | 0.34 | −0.05 | 0.72 | 2.00% |
| Chou and Hung (2021) [34] | 0.09 | −0.56 | 0.73 | 1.43% |
| Coughlan et al. (2022) [28] | 1.02 | 0.65 | 1.40 | 2.02% |
| Deringer et al. (2020) [35] | 0.27 | −0.19 | 0.73 | 1.81% |
| Djernis et al. (2021) [36] | 0.27 | −0.17 | 0.72 | 1.84% |
| Down et al. (2021) [37] | 0.49 | 0.21 | 0.78 | 2.30% |
| Hamann and Ivtzan (2016) [38] | 0.28 | −0.06 | 0.62 | 2.13% |
| Johnson-Pynn et al. (2014)a [39] | 3.40 | 2.59 | 4.21 | 0.97% |
| Johnson-Pynn et al. (2014)b [39] | −0.02 | −0.33 | 0.28 | 2.24% |
| Keenan et al. (2021) [40] | 2.36 | 1.81 | 2.92 | 1.49% |
| Lim et al. (2020) [41] | 0.49 | 0.20 | 0.79 | 2.27% |
| Lumber et al. (2017)a [5] | 0.54 | 0.10 | 0.98 | 1.85% |
| Lumber et al. (2017)b [5] | 0.39 | −0.04 | 0.81 | 1.90% |
| Macaulay et al. (2022)a [42] | 0.21 | −0.08 | 0.49 | 2.30% |
| Macaulay et al. (2022)b [42] | 0.22 | −0.05 | 0.49 | 2.35% |
| Macaulay et al. (2022)c [42] | 0.17 | −0.09 | 0.44 | 2.37% |
| Macaulay et al. (2022)d [42] | 0.15 | −0.13 | 0.42 | 2.35% |
| McEwan et al. (2019) [14] | 0.22 | 0.09 | 0.35 | 2.72% |
| McEwan et al. (2021a)a [43] | 0.39 | −0.09 | 0.87 | 1.75% |
| McEwan et al. (2021a)b [43] | 0.56 | 0.09 | 1.03 | 1.76% |
| McEwan et al. (2021b) [44] | 0.27 | −0.12 | 0.65 | 2.01% |
| Muneghina et al. (2021) [15] | 0.75 | 0.38 | 1.12 | 2.03% |
| Nisbet and Zelenski (2011) [45] | 0.11 | −0.30 | 0.52 | 1.94% |
| Passmore et al. (2022) [46] | 0.10 | −0.31 | 0.51 | 1.94% |
| Ray et al. (2021) [57] | 0.14 | −0.15 | 0.43 | 2.30% |
| Richardson and McEwan (2018) [47] | 0.30 | 0.19 | 0.42 | 2.76% |
| Richardson and Sheffield (2017) [48] | 0.29 | 0.08 | 0.49 | 2.53% |
| Richardson et al. (2016) [49] | 0.29 | 0.19 | 0.40 | 2.77% |
| Richardson et al. (2018) [50] | 0.44 | 0.32 | 0.57 | 2.74% |
| Rogerson et al. (2020)a [51] | 0.79 | 0.43 | 1.15 | 2.07% |
| Rogerson et al. (2020)b [51] | 0.74 | 0.39 | 1.10 | 2.08% |
| Schultz and Tabanico (2007) [52] | 0.16 | 0.02 | 0.31 | 2.70% |
| Sneed et al. (2021)a [58] | 0.30 | −0.11 | 0.71 | 1.94% |
| Sneed et al. (2021)b [58] | 0.48 | 0.08 | 0.89 | 1.94% |

**Table 3.** *Cont.*

| Study Name (Year) Condition | Hedges' g | Lower CI | Upper CI | Weight |
|---|---|---|---|---|
| Spangenberger et al. (2022)a [59] | 0.17 | −0.38 | 0.71 | 1.61% |
| Spangenberger et al. (2022)b [59] | 0.06 | −0.48 | 0.60 | 1.62% |
| Unsworth et al. (2016) [53] | 1.56 | 1.26 | 1.86 | 2.24% |
| Warber et al. (2015) [54] | 0.42 | 0.07 | 0.78 | 2.09% |
| Yeo et al. (2020)a [60] | 0.64 | 0.24 | 1.03 | 1.97% |
| Yeo et al. (2020)b [60] | 0.84 | 0.44 | 1.24 | 1.95% |
| Yeo et al. (2020)c [60] | 0.69 | 0.29 | 1.09 | 1.95% |

In subgroup analyses, neither the quality of engagement ($p = 0.81$), nor the type of contact ($p = 0.73$), nor the timing of intervention ($p = 0.28$) were significant predictors of effect size for nature connectedness. Results were scrutinised for publication bias; Rosenthal's failsafe-N was 6628, indicating that should many additional relevant studies with null results be added, the overall effect size would likely remain significant. However, the funnel plot for the effect sizes appeared to be asymmetric, suggesting there may be evidence of publication bias (see Figure 2). In addition, both Begg and Mazumdar adjusted rank correlation ($\tau = 0.19$, $p = 0.05$) and Egger's intercept (Intercept = 1.71, t = 2.42, $p = 0.02$) were significant, which indicates possible evidence of a publication bias for these data. The impact of the publication bias was assessed by estimating the effect if the bias was absent, by using the Duval and Tweedie trim and fill method [63]. This method removes the most extreme effect sizes from the funnel plot and then re-calculates the effect size to make the funnel plots more symmetrical around the new suggested effect size [63]. Nine data points were imputed on the left of the funnel plot. The adjusted effect $g = 0.30$ [95% CI 0.26, 0.33] $p < 0.01$ was smaller but still represents a significant, positive effect. Relatedly, sensitivity analyses by deleting each study in turn from the analysis [64], indicated that Keenan et al. [40] was a notable outlier; omitting that study reduced the effect, g = 0.42 [95% CI 0.30, 0.54], but still represents a significant, positive effect ($p = 0.001$).

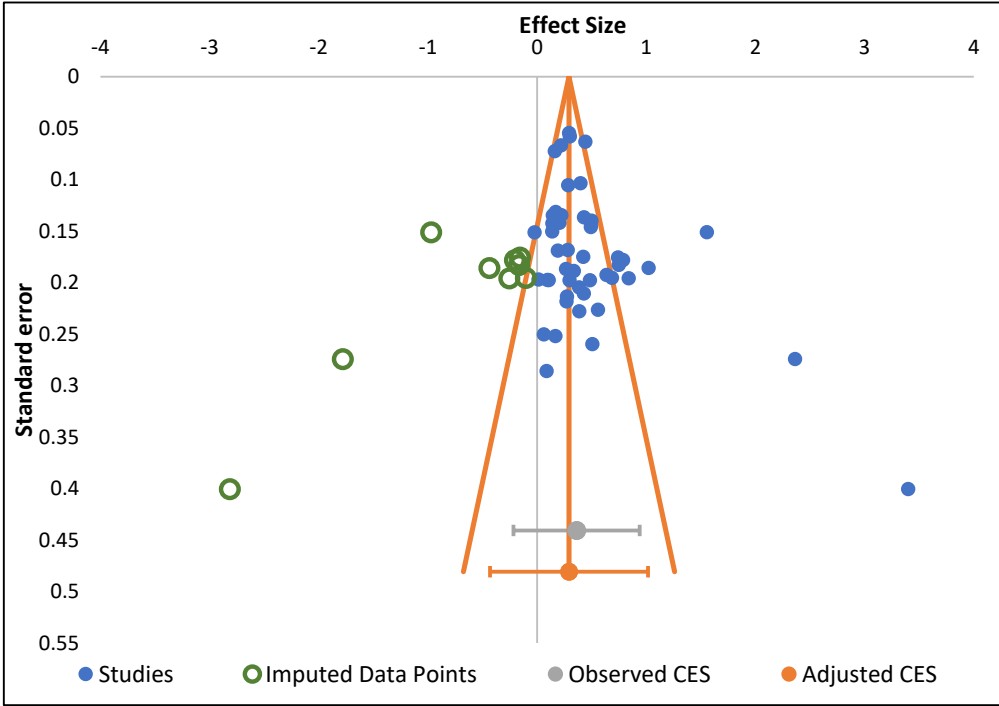

**Figure 2.** Funnel Plot of Pre-Post Effects of Intervention, along with Trim and Fill imputed data.

### 3.2.2. Follow-Up Effects of Interventions

The total sample size for interventions assessing nature connectedness pre-intervention and at follow-up was 1259 (range 10–308) from 11 studies with 12 effects. Table 4 displays the effects of each intervention and Figure 3 shows the forest plot. There was a medium positive mean effect, g = 0.51 [95% CI 0.12, 0.89], which was significant (*p* = 0.004). Heterogeneity of effects was significant (Q = 78.95, *p* = 0.001) and inconsistency was high (I$^2$ = 86.07%).

**Table 4.** Follow-Up Effects and Confidence Intervals of Interventions along with Weighting.

| Study Name (Year) Condition | Hedges' g | Lower CI | Upper CI | Weight |
|:---:|:---:|:---:|:---:|:---:|
| Choe et al. (2020a) [33] | 0.44 | 0.00 | 0.87 | 7.11% |
| Choe et al. (2020b)a [56] | 0.38 | 0.03 | 0.74 | 8.11% |
| Choe et al. (2020b)b [56] | 0.42 | 0.02 | 0.81 | 7.62% |
| Chou and Hung (2021) [34] | 0.60 | −0.11 | 1.32 | 4.82% |
| Djernis et al. (2021) [36] | 0.36 | −0.09 | 0.81 | 6.97% |
| Keenan et al. (2019) [40] | 3.16 | 2.47 | 3.86 | 4.32% |
| McEwan et al. (2019) [14] | 0.19 | 0.00 | 0.37 | 10.28% |
| Muneghina et al. (2021) [15] | 0.75 | 0.38 | 1.12 | 7.86% |
| Richardson and McEwan (2018) [47] | 0.30 | 0.19 | 0.42 | 11.03% |
| Richardson and Sheffield (2017) [48] | 0.29 | 0.08 | 0.49 | 9.99% |
| Richardson et al. (2016) [49] | 0.29 | 0.17 | 0.42 | 10.96% |
| Richardson et al. (2018) [50] | 0.44 | 0.32 | 0.57 | 10.94% |

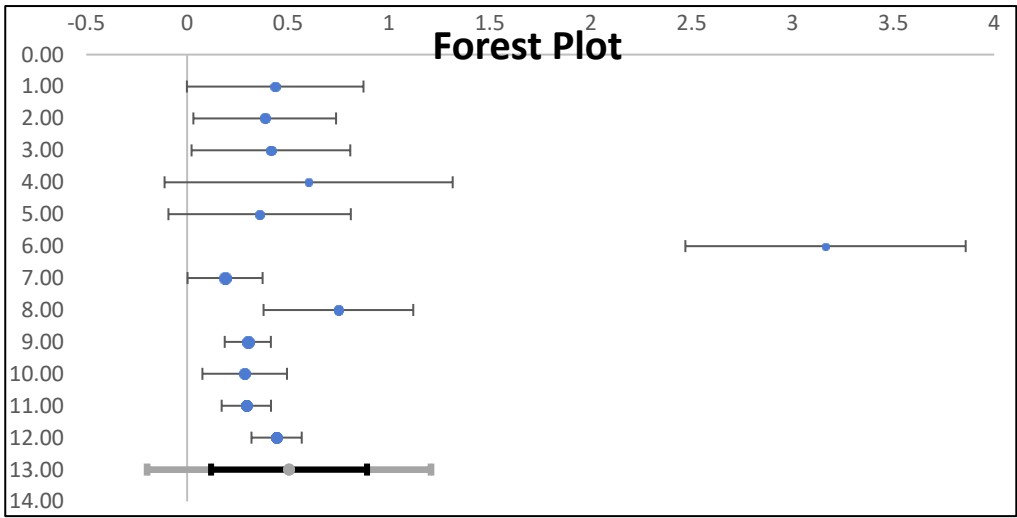

**Figure 3.** Forest Plot of Pre-Follow-Up Effects and Confidence Intervals.

In subgroup analyses, neither the quality of engagement (*p* = 0.33), nor the type of contact (*p* = 0.97) were significant predictors of effect size for nature connectedness; length of intervention was not examined due to the small number of studies in the once only and residential categories.

Results were scrutinised for publication bias; Rosenthal's failsafe-N was 215, indicating that should many additional relevant studies with null results be added, the overall effect size would likely remain significant. However, the funnel plot for the effect sizes appeared to be asymmetric, suggesting evidence of publication bias (see Figure 4). In addition, Begg and Mazumdar adjusted rank correlation (τ = 0.48, *p* = 0.03) was significant and Egger's intercept (Intercept = 2.62, t = 1.96, *p* = 0.08) was close to significant, which indicates

possible publication bias for these data. The impact of the publication bias was assessed by estimating the effect if the bias was absent, by using Duval and Tweedie's (2000) trim and fill method [63]. Two data points were imputed on the left of the funnel plot. The adjusted effect g = 0.33 [95% CI 0.27, 0.39] *p* < 0.01 was smaller but still represents a significant, positive effect. Relatedly, sensitivity analyses by deleting each study in turn from the analysis [64] indicated that [40] was an outlier; omitting that study reduced the effect, g = 0.35 [95% CI 0.26, 0.43], but still represents a significant, positive effect (*p* = 0.001).

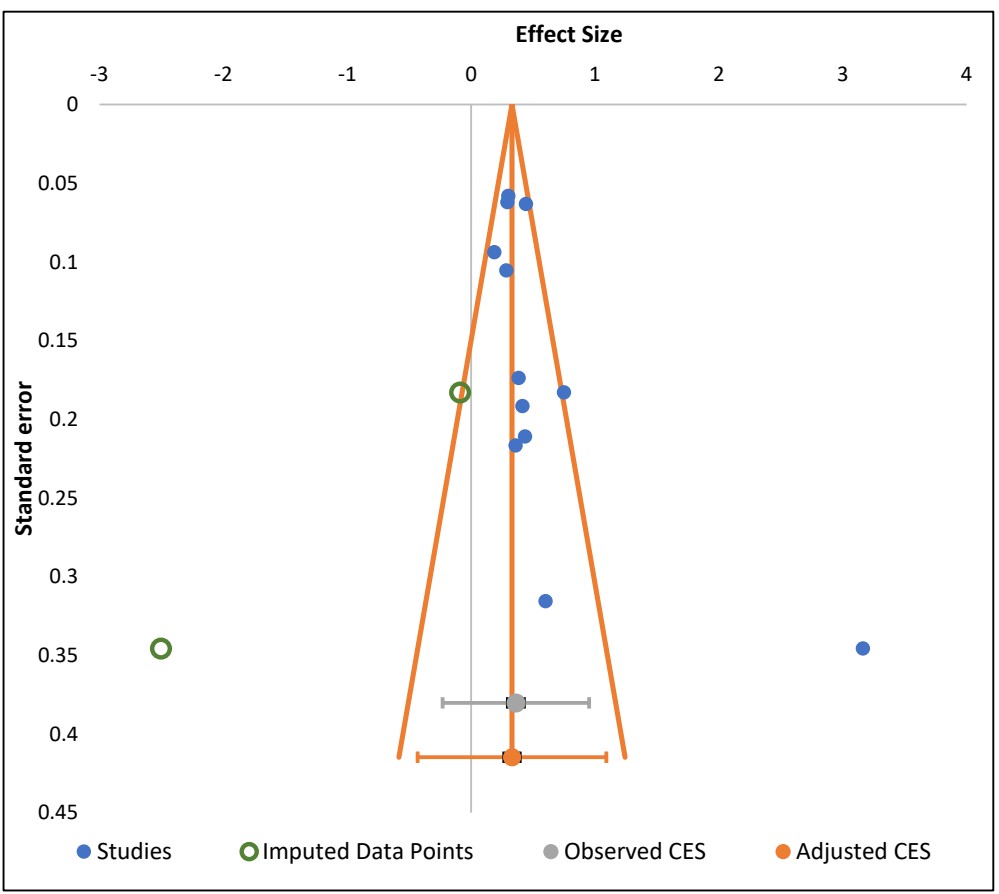

**Figure 4.** Funnel Plot of Pre-Follow-Up Effects of Intervention, along with Trim and Fill imputed data.

## 4. Discussion

This review and meta-analysis examined approaches to improving nature connectedness. Thirty-six studies involving 2855 participants satisfied the eligibility criteria. In terms of short-term improvements in nature connectedness, the interventions had a significant medium positive effect, but neither the type of contact (indirect v direct), quality of engagement (passive v active) or timing of the intervention were significant predictors of effect size for nature connectedness. When considering follow-up data, the 12 studies involving 1259 participants showed significant medium positive effect showing sustained improvements in nature connectedness are possible. Interestingly the baseline to follow-up increases were of a similar magnitude to those measured pre and post. Again, neither the type of contact nor engagement were significant predictors of effect size. Timing of intervention was not examined as all but one study [36] measuring follow-up involved repeated practices.

The results suggest there is clear evidence that contact and engagement with nature can increase nature connectedness, and that engaging in nature connection practices on a regular, daily, or weekly basis can lead to sustained increases in nature connectedness. Given the global calls for improving the human–nature relationship and the positive links between nature connectedness and both human and nature's wellbeing, these findings

are important. Of particular importance is the evidence for sustained increases in nature connectedness, indicating the potential power of carefully designed interventions to create enduring shifts in how people relate to nature.

These findings support those of Barragan-Jason et al. [17] who similarly concluded that manipulations involving nature contact can increase nature connectedness. Notably, we identified 21 studies published after the end of Barragan-Jason's search period in 2020 revealing the rapid growth of research in this area and justifying an updated review. As well as examining the new studies, our focus on adult populations, and exclusion of studies that did not involve direct or indirect contact with nature (for instance, studies examining the effect of university courses on nature connection, such as [65], allowed for a less heterogenous sample and opportunities to examine the quality of engagement and nature of the manipulation more closely. While we found no significant difference between active or passive engagement with nature, or between single, repeated, and residential engagement activities, the meta-analysis helps to clarify distinct qualities of manipulations that are important in future research and development of interventions. To this end, the analysis takes steps towards meeting Lengieza and Swim's [18] calls for greater exploration of the relative impacts of different forms of engagement with nature. Specifically, the sustained benefits observed in the reviewed studies were obtained from studies that included repeated interventions that involved noticing nature or practising mindfulness or engaging with nature. These approaches align with pathways to nature connectedness proposed by Lumber et al. [5] and tested explicitly in some studies.

We have attempted to characterise studies in relation to the quality of the nature contact, by identifying whether participants were explicitly asked to engage psychologically with nature which we coded as 'active engagement', in contrast to passive engagement which is essentially nature contact. Coding engagement for residential camps is particularly difficult as publications are not able to capture the full range of activities participants engage in, so this categorisation should be taken with caution. Of course, we cannot know how all participants in any study actually engaged with nature. Research shows that those who are more connected to nature are more attentive to and emotionally responsive to nature in a range of environments, so some participants in studies asking people simply to walk in nature for ten minutes may well have engaged actively. Conversely, some participants who were asked to engage with their senses may have failed to do so and there may be a lot of variation in the extent or depth of people's ability to do this. These categories suffice, for now, as an initial attempt to organise the research and help set the context for identifying future research directions. They also reveal the importance of authors clearly describing the instructions given to participants and the need for more empirical examination of *how* participants engage with nature independently of any research protocol. More broadly, this highlights the need to develop terminology and conceptualisations around nature engagement.

In the meta-analyses, we chose to exclude studies that do not report pre- and post-scores and focus on studies that examined the magnitude of changes associated with the interventions. Thus, we excluded studies that only report results of treatment comparisons and did not take comparator groups into account due to the wide variety of comparisons made. The lack of comparator groups in the studies included in the meta-analysis (6 of the 12 follow-up studies included a comparator condition) may suggest the effects calculated are over-estimations of benefits of the intervention (although the mean effect in the available comparator conditions was small and non-significant (g = −0.07 [95% CI −0.57, 0.44], *p* = 0.75; see Appendix A). Regardless, the associated review offers only a partial picture of research in the field.

### 4.1. Recommendations for Research

4.1.1. Examine the Impacts of a Wider Range of Nature Engagement Activities

Our analysis shows that research has focused on a rather limited range of one-off or repeated nature contact and engagement activities, such as walking in nature [34,36], meditation



and mindfulness in natural settings [33,53,56], looking at nature images and video [29,55,59,60], appreciating nature [14,38,40,44], and sensory exploration of nature [5,32,43,48]. While there is clear evidence of the efficacy of these practices in growing nature connectedness, additional research is needed to explore the impact of a wider range of activities; activities designed to activate pathways to nature connectedness [5] such as with the *30 Days Wild* intervention [48]. Studies that identify and assess different ways of engaging with nature can help develop the toolbox for individuals and organisations wanting to connect people with nature, and the integration of nature engagement more generally into design, education, policy, and practice. For example, while there are many studies involving walking in nature, there has been little exploration of the impact of sitting with nature. Experimental studies of arts-based engagement with nature are also needed (see [66]). Another area where there is surprisingly little research is the impact of taking part in citizen science activities on connection with nature.

The growing application of the science of nature connectedness by conservation organisations and green social prescribing schemes to promote closer human–nature relationships means that there are an increasing number of activities and interventions being developed, and an increasing need for more evidence-based activities. Empirical research can help to explore the effectiveness of these activities, while also shedding light on what works best.

### 4.1.2. Identify Factors That Result in Biggest and Most Sustained Increases in Nature Connection

We made a distinction between passive and active engagement with nature to categorise the studies, based on what participants were asked to do. There is a need for further research to examine this distinction more closely, and to develop understanding of the different types of active engagement with nature. Carefully designed studies that aim to identify, isolate and test ways of engaging with nature are vital for understanding the most effective pathways to nature connectedness, and the design of interventions.

There are also many open questions as to the impact of other factors on nature connectedness, for instance, what is the effect of being with other people while undertaking nature connection activities? Does social engagement enhance or decrease the impact of nature contact? Another area in need of research is the role of places, objects, and resources in facilitating nature connection—can spending time in spaces designed to connect people with nature promote feelings of closeness? What is the relative impact of the quality of a space compared to people's psychological engagement with nature in that space? A broad urban-nature distinction has been made, but much more needs to be done to examine variations in the quality of natural spaces. What is the effect of different prompts for nature engagement? Most studies reviewed involved verbal or written instructions for activities in nature, while two involved prompts delivered by a smartphone app [14,44]. Additional research is needed to explore the effect of prompts within natural spaces (e.g., a sign or image along a nature walk to promote nature noticing; or an artwork that invites sensory engagement).

Most research has examined the impact of activities on general populations, with very few studies on interventions for more specific groups of people, such as those with mental health or physical health differences—exceptions being two studies [14,40] involving participants with clinically significant mental health conditions. Similarly, little research has explored the impact of individual differences on the relationship between nature contact and engagement and increased nature connection. While trait mindfulness was explored as a mediating variable in some studies, a broader range of personality variables may play a role in the effects of nature on feelings of connectedness. There is a need to bring together findings from cross-sectional studies to identify potential barriers to nature connectedness and identify new approaches for nature engagement. Different activities and contexts may work better for some groups than others. What works for people high in trait mindfulness may not work so well for those less mindful.

### 4.1.3. Design and Test Practices for Growing Sustained Nature Connection

Research indicates a small to medium effect for one-off nature contact and engagement activities, suggesting that nature contact and engagement can cause an immediate increase in feelings of nature connectedness. There is little to no evidence to suggest that brief one-off activities have any impact on nature connection over the medium to long-term as few such studies have included a follow-up. However, there is evidence that engaging in nature connection practices on a regular, daily, or weekly basis leads to sustained increases in nature connection. It is these enduring changes in how people feel about their relationship with nature that are important for meaningful impact on pro-environmental behaviour and wellbeing. The gold standard for nature connectedness interventions is the development of practices for routine active psychological engagement with nature that establish lasting feelings of closeness.

One avenue to explore is whether activities found to lead to short-term spikes in nature connectedness may serve as the basis for sustained nature connectedness, if the activity is repeated on a regular basis. The studies that did show sustained effects tended to involve close psychological engagement with nature, involving nature noticing, appreciation, and activation of the pathways to nature connection [5]. Is it the active engagement with nature that matters for lasting impact (as suggested by cross-sectional research [16], or is the act of creating habitual ways of being with nature the key? There is a lot of scope for development of additional activities that aim to activate pathways to lasting nature connectedness, and research exploring the feasibility and efficacy of these. Of key importance, however, is identification of factors that make an intervention appealing to people to try in the first place, and to maintain as regular practice.

### 4.2. Recommendations for Practice

### 4.2.1. Engage People with Nature

The research shows that asking people to engage with nature increases feelings of nature connectedness. While additional research is needed to deepen understanding of this relationship and support development of interventions for lasting nature connection, evidence is already in place for real-world application. There is an urgent need to expand and intensify the delivery of initiatives that seek to increase nature connectedness. In the United Kingdom, green social prescribing is rapidly expanding with nature engagement being promoted by GPs and other health professionals to support people's physical and mental wellbeing [67]. For population-level shifts in the human–nature relationship, nature-connecting activities can be incorporated into public health strategies and initiatives. Campaigns for nature engagement should sit alongside campaigns for physical activity or healthy eating, to prevent and treat the symptoms of a broken relationship with nature.

Nature engagement activities are widely used by nature and conservation-based organisations looking to support visitor experience and learning. However, as indicated by the studies involving urban nature or indirect nature experiences, wildlife-rich nature reserves or forests are not a pre-requisite for growing nature connection. While they may certainly offer greater opportunities for connection and greater biodiversity increases people's feelings of closeness with nature [68,69], nature engagement can happen almost anywhere. What is needed are opportunities, prompts and invitations for people to engage with nature, and these can be delivered across the whole spectrum of sectors and organisations. For example, schools can incorporate nature engagement for children and young people into teaching and wellbeing initiatives, businesses can promote nature connection amongst employees through a programme of engagement activities for combined environmental and social sustainability, and community organisations can facilitate nature connecting activities for local residents.

### 4.2.2. Create Conditions for Nature Connection

While nature connectedness can be increased by engaging with the simple activities and practices tested in the experimental research, steps can be taken to maximise opportu-

nities for these to be carried out. Key to this is the provision and recovery of nature spaces to facilitate walks, sensory engagement, and appreciation of nature's beauty and wonder. On a national and local scale this can be achieved through centralised decisions about land use and development and urban planning, recognising the value of nature engagement opportunities and ensuring that nature-rich spaces are available and accessible to all, and support for design that has nature connection as a key target. On a smaller scale, spaces inside and outside businesses and homes can be used to create opportunities to notice and appreciate nature.

The research shows that even short moments of engaging with nature can increase feelings of connection. Institutions and organisations can facilitate opportunities for people to take such moments, for instance, by identifying places and times where people spend time waiting, or by promoting nature breaks during the workday. The positive effects of even short periods of time looking at nature photos and videos suggests the value of providing such material in workplaces, waiting rooms and public spaces.

### 4.2.3. Encourage Repeated Nature Engagement Activities

Sustained increases in nature connectedness were observed when people were invited to engage with nature on a daily or more regular basis. While more research is needed to develop and test interventions for sustained nature connectedness, there is already sufficient evidence to support ongoing development of programmes and practices of regular nature engagement. Campaigns like the Wildlife Trusts' *30 Days Wild* offer population-level examples, while green social prescribing programmes (e.g., [67]) can invite people to engage with nature on a regular basis. Where one-off experiences are offered, people should be encouraged to repeat the experience, or elements of it, to reap the benefits of regular nature engagement. Campaigns, 'challenges,' and the development of personal practices that involve regular nature contact and engagement can help establish new ways of being with nature. This integration of nature into everyday life, with increased awareness and appreciation of the natural world, is essential for lasting changes in the human–nature relationship.

### 5. Conclusions

Targeting sustained improvements in nature connectedness can help address the global calls for a new relationship with nature required for a sustainable future. The analysis above confirms that carefully designed interventions can deliver sustained increases in nature connectedness. Those sustained benefits typically involve repeated interventions that create the conditions for people to engage with nature. When people are prompted to engage with nature regularly, they develop a closer relationship with it. Although simple, this finding is important. Many people do not 'notice nature' and urban residents often spend only a few minutes in green spaces each day [69], with the biodiversity of those green spaces being important for the benefits they bring and opportunities they provide for repeated noticing and engagement with nature [68,70]. While ultimately there needs to be widespread recognition of the interconnectedness of nature's and people's wellbeing, the mechanism for achieving either is the same—help people feel closer to nature. Efforts to do this should be central to any and all organisations or policies that set out to improve the wellbeing of humans, nature or both. Interventions could range from programmes focussed on individuals to those creating the conditions for nature connectedness through considering macro factors such as the design of urban areas and land use [71]. Although options for fostering connectedness are available now, the range is limited and further research is required into ways of engaging people with nature, and the integration of nature engagement into policy areas such as housing, urban planning, and education.

**Author Contributions:** Conceptualization, C.W.B., D.S. and M.R.; methodology, C.W.B., D.S. and M.R.; formal analysis, D.S.; writing—original draft preparation, C.W.B., D.S. and M.R.; writing—review and editing, C.W.B., D.S. and M.R. All authors have read and agreed to the published version of the manuscript.

**Funding:** This research received no external funding.

**Institutional Review Board Statement:** Not applicable.

**Informed Consent Statement:** Not applicable.

**Data Availability Statement:** Data will be made available at UDORA, the University of Derby open access repository.

**Conflicts of Interest:** The authors declare no conflict of interest.

**Appendix A**

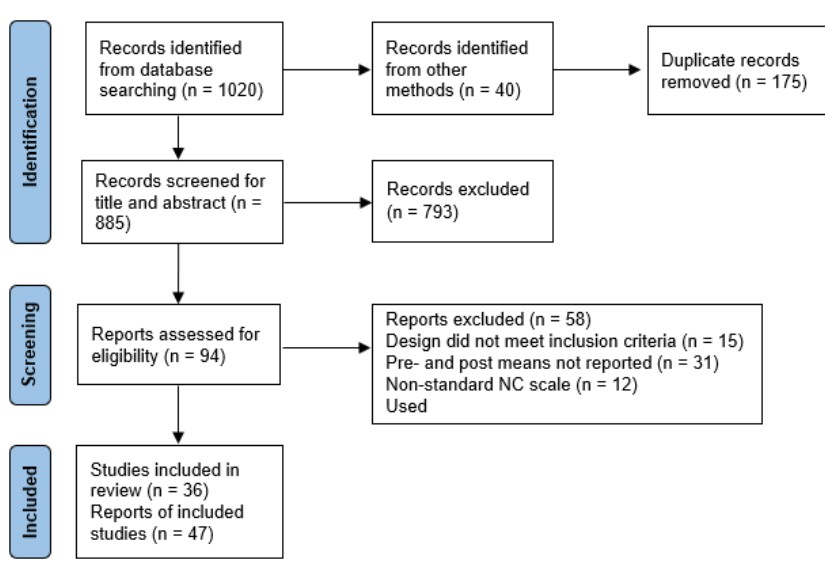

**Figure A1.** PRISMA Flowchart.

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
