# Peer review of "Improving Nature Connectedness in Adults: A Meta-Analysis, Review and Agenda"

_sustainability, doi:10.3390/su141912494_

Round 1

Reviewer 1 Report

This study is devoted to the analysis of literature in the field of the relationship between people and nature. This study is conceptual. The methodology corresponds to the set goal. the author's idea is clearly expressed, and the ways to achieve it, too. The abstract corresponds to the content of the article. However, the authors need to correct the title: " Improving natural connectivity in adults: A meta-analysis, review and agenda (Review)Improving nature connectivity in adults: A meta-analysis, review and agenda". The authors should still choose one. The list of keywords can be expanded. The inclusion of the Recommendations for Practice section is very positive. This section significantly differs from this study from similar ones. However, the Conclusion section should be expanded. It is necessary to include the main provisions of the results in this section.

Author Response

Thank you for your helpful review.

Title – “nature connectedness” is the correct term. We recognise that connectivity has on occasion been used but following Mayer and Frantz (2004) we prefer nature connectedness and think most readers will recognise this term.

Mayer, F. S., & Frantz, C. M. (2004). The connectedness to nature scale: A measure of individuals' feeling in community with nature. Journal of Environmental Psychology, 24(4), 503–515.

  – (Review) is not part of the title.

We have added keywords.

The Conclusion section had been expanded to include the main provisions of the results, as requested.

Reviewer 2 Report

Thank you for the opportunity to review your manuscript reporting an impressive meta-analysis of effects generated across studies measuring nature connectedness and specifically comparing direct vs. indirect contact, active vs. passive engagement, and single session vs. repeated practice vs. residential timing of engagement.  That fact that no effect size differences were observed speaks to the importance of nature connectedness itself rather than the form in which it is offered. The agenda for future research and practice will be of great use to researchers, practitioners and urban planners as a guide to developing new opportunities or to enhance those already in place.  

Other than my appreciation for the quality of the research reported, I have only some very minor typographic issues to note.

Line 4. Richardson misspelled

Line 62. Word missing; “greater understanding how nature connectedness”

Lines 74-76. Sentence incomplete: longer term what? “They also found that while both short and longer interventions enhanced nature connectedness immediately post intervention only longer interventions resulted in longer term (≥2 weeks).”

Line 146. Word missing; “papers the authors’ collections”

Tables 1 & 2. Don’t fit onto the page

Lines 216-219. Either/or incomplete. “Five studies invited participants to either carry out nature-based activities in their own time [32-35], while four invited participants to engage with nature in particular ways (i.e., appreciating and noticing it) during their daily lives without asking them to spend any extra time outside [14,36-219 38].”

Line 272. Inappropriate capitalization; “the course of 30 Days”

Table 3. Column alignment needs adjustment

Lines 365-379. Formatting has gone awry

Line 516. Redundant parenthesis. “needed (see [65])).”

Line 568. Redundant numeral. “brief one-off activities have any l impact”

Line 573. Redundant full stop. ”behaviour and wellbeing. . The”

Author Response

Many thanks for your review. All the minor typographic issues have been corrected.

Reviewer 3 Report

This paper conducted a meta-analysis of nature connectedness in adult populations and explored how experimental manipulations and field interventions could improve an individual’s sense of nature connectedness. But I couldn’t find the definitions of nature connectedness. The search methods were clearly and well-aligned with similar literature on the topic. The results are logical and well explained in the text, while it might be more concise by moving some figures to the supplemental materials and just mentioning major results to improve readability and interpretation. In addition, table 1 & table 2 already provided sufficient information about the “activity”, it’s not necessary to describe the detailed results among the three study dimensions (Line 208- Line 285).

Author Response

Thank you for your careful review.

A definition of nature connectedness is provided on line 46-48

Line 208-285 is a large section – as the other two reviewers are very positive and have not noted the need for substantial revisions to this section we feel it is best not to make major changes. Moreover, we believe readers will find the summaries useful.